# Delphi Method to Achieve Clinical Consensus for a BPMN Representation of the Central Venous Access Placement for Training Purposes

**DOI:** 10.3390/ijerph17113889

**Published:** 2020-05-30

**Authors:** Rene de la Fuente, Ricardo Fuentes, Jorge Munoz-Gama, Jorge Dagnino, Marcos Sepúlveda

**Affiliations:** 1Department of Anesthesiology, School of Medicine, Pontificia Universidad Católica de Chile, Santiago 8331150, Chile; rdelafue@med.puc.cl (R.d.l.F.); jdagnino@med.puc.cl (J.D.); 2Department of Computer Science, School of Engineering, Pontificia Universidad Católica de Chile, Santiago 7820436, Chile; jmun@uc.cl (J.M.-G.); marcos@ing.puc.cl (M.S.)

**Keywords:** business process modelling notation, surgical procedures, procedural skills training, central venous catheter, cognitive task analysis

## Abstract

Proper teaching of the technical skills necessary to perform a medical procedure begins with its breakdown into its constituent steps. Currently available methodologies require substantial resources and their results may be biased. Therefore, it is difficult to generate the necessary breakdown capable of supporting a procedural curriculum. The aim of our work was to breakdown the steps required for ultrasound guided Central Venous Catheter (CVC) placement and represent this procedure graphically using the standard BPMN notation. Methods: We performed the first breakdown based on the activities defined in validated evaluation checklists, which were then graphically represented in BPMN. In order to establish clinical consensus, we used the Delphi method by conducting an online survey in which experts were asked to score the suitability of the proposed activities and eventually propose new activities. Results: Surveys were answered by 13 experts from three medical specialties and eight different institutions in two rounds. The final model included 28 activities proposed in the initial model and four new activities proposed by the experts; seven activities from the initial model were excluded. Conclusions: The proposed methodology proved to be simple and effective, generating a graphic representation to represent activities, decision points, and alternative paths. This approach is complementary to more classical representations for the development of a solid knowledge base that allows the standardization of medical procedures for teaching purposes.

## 1. Introduction

The development of procedural skills is an essential component in the process of training physicians of many medical specialties. It has been shown that superior technical performance positively affects patient outcomes [1] and that the presence of technical deficiencies is the most important factor associated with operator errors in hospitalized patients [2]. Procedural instruction using simulation before contact with patient has proven to be a useful tool in the development of these skills [3,4]. A first stage, before procedure training using simulation, is to break down the procedure into observable and ordered steps that allow a clear and thorough description, including those points where decisions are made [5,6]. Cognitive Task Analysis (CTA) has been used to make explicit the knowledge that experts have about a procedure. When teaching, expert may omit up to 70% of the knowledge necessary to perform a procedure [6,7]; apprentices must then fill in these gaps individually using a trial and error approach, causing difficulties in the learning process. There are multiple methodologies based on CTA, with different mechanisms to make knowledge explicit, to analyze it and to represent it [8,9]. The drawbacks of these methodologies are the amount of resources required-hours of interview and analysis, and the need for trained people [6,8], and the fact that, for the same procedure, the breakdown may differ depending on the specialty of the experts [10]. This limitation makes it difficult to generate a procedural curriculum that can be readily accessed by instructors and apprentices. Business Process Model and Notation (BPMN) [11] is the de facto standard used for designing and modelling business processes. Business process models are a key element of the so-called BPM lifecycle, where processes are continuously identified, modelled, analysed, improved, and monitored [12]. The use of BPMN in healthcare has proven to be a versatile way to efficiently represent surgical processes, such as clinical pathways, of different complexity [13,14], being a tool that can be used as the basis for institutional performance protocolization and to improve adherence to good practices [13,15,16]. Furthermore, it is important to consider the modelling as a fundamental part of Business Process Management oriented towards continual improvement as part of quality programs. Through the use of standardized elements with a clear syntax, it achieves a highly expressive graphic representation of the structural and functional aspects of a healthcare process [17], and is easily understood by health personnel [13,14]. Central venous catheter (CVC) placement is a critical medical competency, particularly for critical care physician and anesthesiologists. The use of ultrasound guidance for CVC placement is strongly recommended when the anesthesia provider has training and experience with the technique [18]. Simulation training has become an effective option to improve providers’ performance [19] and reduce clinical risks [20,21]. The aim of our work was to break down the steps required for ultrasound guided CVC placement and represent this procedure graphically using the standard BPMN notation.

## 2. Methods

### 2.1. Development of an Initial Process Model of Ultrasound Guided CVC Placement

After approval by the institutional ethics committee (Id: 16-194), we began by identifying the steps required for ultrasound guided CVC placement. We used the activities defined in the checklists proposed by Barsuk et al. [4] to evaluate the installation of dialysis catheters, which fulfills the requirement of representing all skills, ensuring the inclusion not only of procedural aspects related to motor skills, but also those related to safety, infection control, and preparation. Use of this checklist in the training context has been validated to demonstrate the achievement of competences at a sufficient level to have a favorable impact on clinical results [20,22]. We extended this description with the activities included in the checklist developed by Nguyen et al. [23], since it describes the necessary steps to correctly include the use of ultrasonography in the procedure. An ultrasound guide has been shown to have a positive impact on success rates and to reduce morbidity in this procedure [24,25,26]. We then complemented them with the activities defined in the reviews of McKinley et al. [27] for checklists and Ma et al. [28] for global scales in the installation of central venous accesses. With these activities and using basic elements of BPMN (Figure 1), we first developed the ultrasound guided CVC placement model as a list of tasks (activities) ordered by a sequence flow (arrow connecting two consecutive tasks) and inclusive (and/or for complementary activities), exclusive (Yes/No for exclusive activities), or parallel gateways (used to control splitting and merging of sequence flows). Some tasks contain qualitative attributes about the representation as text annotations. This initial model of the procedure was then evaluated by three anesthesiologists, part of the team of the Pontificia Universidad Católica Clinical Simulation Center, who use checklists, global scales, and measurements with the Imperial College Surgical Assessment Device (ICSAD), for research into procedural skills. These activities resulted in a first complete graphic process model of the procedure.

### 2.2. Consensus through the Delphi Method

To reach a model that can be used in different healthcare centers, avoiding the biases of specialty or local practices, we assembled a Delphi panel. The Delphi method has proven to be effective in many disciplines in order to obtain consensus among experts on a given subject [29,30]. Its main characteristic is the anonymous interaction between experts who, having access to the answers of the other participants, can modify their own answers in successive and controlled rounds. The process concludes when the answers represent the opinion of the majority of the group [31]. We used the method suggested by Okoli and Pawloski [32] to select our experts including, as such, physicians with at least a five-year experience in ultrasound guided CVC placement with a mean of five to seven catheter insertions per month, and with at least one of the following attributes: (a) Person locally in charge for the procedure, (b) Participation in national guidelines development, (c) Instructor for the procedure in the Chilean Society of Anesthesiology workshops, (d) Head of a department where CVCs are placed. A group of 16 national experts were thus selected and invited to participate, via an email describing aims and procedures; following acceptance, a first round survey (Appendix A) was sent; this was designed using the online platform Surveymonkey^®^ (www.surveymonkey.com), structured in five parts:Display of the proposed BPMN model (Appendix A).Personal data acquisition: specialty, work area, contact data.Set of structured questions, where all the activities of the previously defined BPMN model are listed sequentially. Each question describes the activity and asks each expert to weigh the appropriateness of its inclusion in the model, based on a 5-point Likert scale: 1. Under no circumstances should be included, 2. Should not be included, 3. May or may not be included, 4. Should be included, 5. Must be included.Addition of new activities not included in the proposed model, defining the place they should occupy in the model.Proposal of other experts in the ultrasound guided CVC placement, who could participate in the research.

The results of the first survey were consolidated using the percentage of responses for each item on the Likert scale for each of the proposed activities were consolidated in three categories to facilitate their visual perception of each activity; this was done using the percentage of responses for each item on the Likert scale: Not to Include (“Under no circumstances should be included” plus “Should not be included”), Neutral (“May or may not be included”) and Include (“Should be included” plus “Must be included”). In addition, the new suggested activities were analyzed, evaluating if they coincided with activities already included or if they were new activities not considered in the initial model. Finally, an invitation to participate in the study was sent to experts mentioned by the respondents. A second survey (Appendix A) included all the activities of the first survey and the consolidated percentage each of them had obtained on the Likert scale in the first round, plus the new activities proposed by the experts (Appendix A). The experts were again invited to weight each activity, based on the same 5-point Likert scale. The Delphi panel took into account the methodological quality criteria proposed by Diamond et al. [29]: a reproducible selection of participants and the definition of a stopping criterion, a maximum number of rounds, and an exclusion criterion for each item. The criterion for inclusion in the model was predefined as those activities that had a percentage greater than or equal to 75% of the sum of the responses “Should be included” and “Must be included”; this nonrestrictive criterion was adopted so as to ensure consensus without exclusion of activities that would show inter operator variability [29,30]. A numerical value of 1 to 5 was assigned to each of the items on the Likert and the average value was calculated for each question in the survey. Finally, for each question, the percentage change between two successive surveys was obtained by subtracting the average value obtained in the first survey from the average value obtained in the second survey, divided by the average value obtained in the first survey. Notice that the percentage change between two successive surveys can be either positive when the item is considered more relevant, or negative when the item is considered less relevant. An average of the absolute percentage changes in all responses of less than 15% in successive rounds was defined as the stopping criterion for ending the rounds [32], planning a maximum of three rounds if that criterion was not met.

## 3. Results

### 3.1. Experts

Of the 16 invited experts, 12 accepted to participate and completed the two online survey. They proposed four new experts, of which three agreed to participate, but only one answered the survey. In total, 13 experts formed the Delphi panel, representing different specialties, country regions and hospital characteristics; this variance is shown in Table 1.

### 3.2. Model

The first survey proposed 34 activities. Eight experts made at least one proposal to enrich the model. In total, there were 14 proposals, of which:Five were included as new activities in the second survey.Five activities already considered were modified, either by redefining an activity or by adding some qualitative attribute.Four activities were not included because they escaped the scope of the model (e.g., catheter tunneling, or avoiding the patient taking deep breaths).

The second surveys proposed 39 activities: 34 included in the first survey and five new activities. The stopping criterion of the Delphi panel (Table 2) was met in the second round of surveys, which showed an average of the absolute percentage changes in all responses of 3.62% in relation to the responses to the first survey, with a standard deviation of 3.81%. The percentage changes range between −14.58% and 4.25%.

The final model (Figure 2) considers 32 activities. Of the total activities of this model, 27 were included because they met the inclusion criteria, 23 activities of the original model, and four new activities proposed by the experts. Of the activities that did not reach consensus among the experts, seven were excluded and five other activities were handled as follows:Of the three alternatives proposed as activities to verify the position of the vein with ultrasonography before puncture (Anatomical identification, compression, or use of Doppler), none reached the inclusion threshold. When analyzed as a whole, 92% of respondents used at least one of these three alternatives, so it was decided to include all the possibilities in the graphic model.“Check catheter in the vein with ultrasound in short axis” inside the vein lumen, obtained 67% on the inclusion criterion; and “Check catheter in the vein with ultrasound in long axis”, obtained 75%. Therefore, it was decided to include both as variants of the same activity.The activity “Put sterile gel on covered transducer” did not reach the inclusion criterion. However, we considered that the way the item was worded was incorrect, given that the sterile gel is sometimes used on the puncture site to generate an interface that avoids the presence of air and not necessarily on the transducer. This could have caused confusion among the experts. Therefore, we modified its name to “Put sterile gel” and included it as such in the final model.

## 4. Discussion

The main result of this work was the construction of a representation of ultrasound guided CVC placement with the use of BPMN. BPMN allows creating a model of the successive steps necessary to correctly execute a procedure. It is a versatile tool that enables representing non-dichotomous decision points and the existence of cycles, through gateways and sequential flows that return to previous tasks. These characteristics are an advantage over the classic representation based on narrative descriptions or tables, which are usually sequential and unidirectional. The use of BPMN as a notation for the model has the advantage that it uses standardized concepts [11], with a precise syntax, easily understood by both the healthcare team and the other staff, including the Information Technology (IT) department [13]. Despite this, there are limitations that must be considered at the time of their application in the healthcare field, such as the absence of explicit information on the level of clinical evidence on each of the proposed steps, information that is not only useful when using the model in learning situations, but also to guide the executor in decision-making. Recently, Braun et al. have proposed an extension of the BPMN notation that attempts to deal with this limitation [33].

Our modelling proposal includes the use of checklists as a basis for the breakdown. These checklists consist of a list of observable activities or behaviors organized in a consistent manner, which allow an observer to record the presence or absence of these activities in the context of an evaluation [34]. A recent review by Ma et al. identified 25 checklists used to evaluate the installation of central venous accesses in different contexts [35], with a median of 17 items and range between 2 and 63. Starting the breakdown from one or more validated checklists has the advantage of working with instruments that have proven useful in abstracting those activities considered critical when evaluating the achievement of a competence. On the other hand, although these checklists can be considered an abstraction of the procedure, they are not enough to express the critical decision points or the possible existence of alternative paths, which is why these activities were used only as a basis to develop a model in the BPMN notation.

The next step, enriching and validating the breakdown generated through a Delphi panel with experts from different specialties and different types of healthcare institutions (Table 1), allowed us to avoid a frequent problem that needs to be dealt with when attempting to abstract the execution of procedures: the significant variability in its execution that can be determined by patients, skills and experience of the team, and available technologies [36,37]. This is critical when the purpose of the breakdown is its use for the training of residents who will develop professionally in different types of institutions. Thus, the inclusion of experts from different institutions allows the generation of a breakdown applicable to different scenarios without local biases [10]. Surprisingly, only two rounds were necessary to reach consensus, showing a high inclusion rate in the final model of those activities that had been initially proposed; this could be explained by the fact that the first generic model was developed using previously validated instruments, in which unnecessary steps had been depurated, for ultrasound guided CVC placement. Nonetheless, the “skin cut” step before dilatation, present in various checking lists [27,35], was not considered necessary by a majority of the experts.

We propose that this model can be used as input in the training activities of residents for the acquisition of many procedural skills in a simple, without requiring specialized resources, and easy to interpret alternative to those currently in use. Our approach takes into account the work already done for the development of evaluation checklists, assessing how they fit training needs and their quality in representing the competencies involved [27]. The generation of a highly versatile graphic representation that is submitted to a consensus-building process among experts permits the reuse of work already carried out and produces a result in a collaborative way that satisfies several cross-specialty needs, allowing access of trainers and trainees a graphical representation of the necessary steps to successfully achieve an ultrasound guided CVC placement in preclinical training stages.

Curricular weaknesses referring to technical skills are most critical in surgical specialties where a significant percentage of residents feel insecure about facing surgery autonomously on finishing their training [38] and program directors of surgical sub-specialties consider that surgeons who enter their programs have deficiencies in necessary technical skills at the time of beginning their training [39]. In this context, having the breakdown of a procedure into its constitutive activities and its decision points allows an increase in the effectiveness of the training process [5,9,40]. However, the need for specially trained personnel and the time involved in applying the currently available methodologies to provide a pre-graduate or post-graduate program with the total number of broken down procedures necessary to generate a procedural training curriculum can be an excessive burden for most institutions [7,41]. The proposed methodology is a suitable tool to quickly supply brokendown procedures, as a first step towards developing a procedural training curriculum.

Although our approach begins with a model created based on a checklist and then it is extended based on a consensus among local experts, we notice that an equally valid alternative approach is to build this initial model in the opposite direction: use local insights and then extend it. This can be useful in those cases where you want to give greater weight to the local vision or do not have checklists with a sufficient level of detail.

## 5. Limitations

Among the limitations of our study to be considered, the procedure chosen may not be one that most benefits from the use of alternatives to CTA, as a recent study by Yates et al. [42] shows that the level of omissions made by experts when breaking down this procedure is less than 30%. On the other hand, even if BPMN has been applied to many surgical procedures [43], being considered intuitive, easy to understand and useful in different clinical contexts [13,14,44]; the fact that its use is advantageous for training purposes needs to be explicitly evaluated. A good approach to this is to develop a feasibility study, in order to make its comprehensibility for trainers and trainees explicit. Furthermore, it is essential to corroborate the potential of the proposed approach with other procedures of greater complexity or performed less frequently.

Another limitation that needs to be considered is that our model only included the steps necessary for an uncomplicated ultrasound guided CVC placement (happy path), leaving out events such as an arterial puncture or the placement of a catheter in the artery. These complications and their corresponding management are addressed in the American Society of Anesthesiology guidelines [26]. The inclusion of all possible complications can result in a considerable increase in the complexity of the representation, but it may be important to include the most frequent or the potentially most significant complications, this being a training tool. Another possible approach is to have two models, one for the initial stages of training and the other more complex and exhaustive for more advanced stages, thus avoiding cognitively overloading the residents. Our decision to include only national experts could compromise its wider applicability, but its development in realities of limited resources demonstrates that this methodology can be used for the breakdown of steps for ultrasound guided CVC placement or other procedures.

## 6. Conclusions

The proposed methodology allows the breakdown and representation of a procedure common to several surgical and medical specialties in a versatile and highly expressive graphic notation. The use of checklists previously psychometrically validated and of proven impact on clinical outcomes, plus the enrichment of the breakdown through the consensus of experts using a Delphi method, allows its use for training purposes. This methodology would allow the development of a robust knowledge base for a variety of procedures, enabling the standardization of teaching procedures for undergraduate and graduate students, in a simple manner, without requiring specialized resources.

## Figures and Tables

**Figure 1 ijerph-17-03889-f001:**
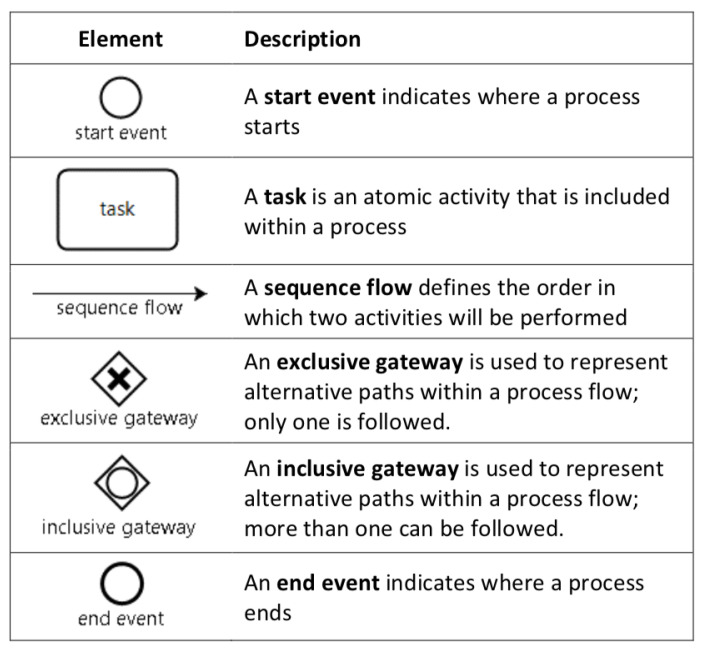
Basic elements of business process model and notation (BPMN).

**Figure 2 ijerph-17-03889-f002:**
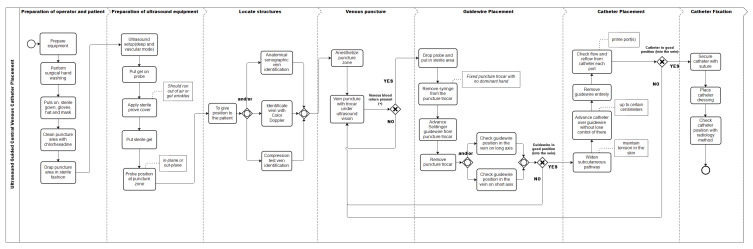
Process model of the central venous access placement with ultrasonography after delphi panel, represented in BPMN.

**Table 1 ijerph-17-03889-t001:** Characteristics of the participants in the Delphi panel.

Characteristics	Description	Number
Experts by Specialty	Anesthesiology	8
	Anesthesiology and Intensive Medicine	4
	Nephrology	1
Experts by unit of work of work	Operating Room	7
	Operating Room and ICU	4
	ICU	1
	Dialysis Unit	1
Experts by type of hospital	Private Hospital	4
	Public Hospital	3
	University Hospital	4
	Military Hospital	2
Expert by geographic location	Capital City	10
	Regional Cities	3

**Table 2 ijerph-17-03889-t002:** Delphi panel results. First survey, second survey and percentage change of each response between the two rounds. Display responses of the first and second round of the Delphi panel.

Proposed Activity	First, Survey			Second Survey			Change
	Not Include %	Neutral %	Include %	Not Include %	Neutral %	Include %	%
Prepare implements	0.0	8.3	91.7	0	0	100	4.25
Perform surgical hand washing	0.0	0.0	100.0	0	0	100	0.00
Get in sterile gown. gloves. hat and mask	0.0	0.0	100.0	0	0	100	−2.08
Clean puncture area with chlorhexidine	0.0	0.0	100.0	0	0	100	2.00
Drape puncture area in sterile fashion	0.0	0.0	100.0	0	0	100	0.00
Ultrasound setup (deep and vascular mode)	0.0	16.7	83.3	0	17	83	−2.08
Put gel in probe	8.3	8.3	83.3	8	17	75	−4.08
Cover sterile probe	0.0	0.0	100.0	0	0	100	4.17
Put sterile gel in covered transducer	16.7	33.3	50.0	17	42	42	−2.17
Probe position at puncture zone	0.0	8.3	91.7	0	17	83	0.00
To give position to the patient ⋆				0	17	83	
Forced contralateral head rotation	16.7	58.3	25.0	25	58	17	−2.17
Do Valsalva maneuver	25.0	58.3	16.7	33	67	0	−8.42
Anatomical vein identification	8.3	33.3	58.3	17	25	58	−2.00
Identify vein with color Doppler	0.0	50.0	50.0	0	83	17	−14.58
Compression test identification	0.0	33.3	66.7	0	33	67	2.17
Anesthetize puncture zone	0.0	25.0	75.0	0	25	75	−2.00
Puncture with fine needle	41.7	33.3	25.0	42	50	8	−8.25
Vein puncture with trocar under ultrasound vision	0.0	8.3	91.7	0	0	100	2.08
Venous blood return +	0.0	16.7	83.3	0	17	83	−2.00
Set probe in sterile area ⋆				0	17	83	
Remove syringe from the puncture trocar	0.0	25.0	75.0	0	17	83	−2.17
Advance Seldinger guidewire through puncture trocar	0.0	8.3	91.7	0	0	100	2.08
Remove puncture trocar ⋆				0	17	83	
Verification of the guide with ultrasound in long axis	0.0	16.7	83.3	0	25	75	−10.33
Verification of the guide with ultrasonography in short axis	0.0	33.3	66.7	0	33	67	−4.25
Guidewire in good position (in the vein)	0.0	0.0	100.0	0	8	92	−4.08
Cut skin	25.0	58.3	16.7	33	58	8	−4.25
Widen subcutaneous pathway	0.0	8.3	91.7	0	0	100	−4.08
Advance catheter over guidewire without losing control	0.0	0.0	100.0	0	0	100	2.17
Remove guidewire entirely	0.0	0.0	100.0	0	0	100	−0.08
Check flow and reflow in each catheter port	0.0	8.3	91.7	0	17	83	−2.17
Check catheter in the vein with ultrasound in long axis	0.0	41.7	58.3	8	50	42	−8.42
Check catheter in the vein with ultrasound in short axis	8.3	41.7	50.0	17	50	33	−8.42
Catheter in good position (in the vein)	0.0	8.3	91.7	0	8	92	−2.17
Secure catheter with knots	8.3	0.0	91.7	0	17	83	−0.08
Secure catheter with other systems ⋆				8	58	33	
Place catheter patches	0.0	8.3	91.7	0	8	92	−0.08
Check catheter position with radiology method ⋆				0	17	83	

Definitions: Not include: percentage of responses “Under no circumstances should be included” or “Should not be included”; Neutral: percentage of responses “May or may not be included”; Include: percentage of responses “Should be included” or “Must be included”. Activities green rows: Activities not included in the final model. Activity Venous blood return: + is Positive. Activities not included in model the first survey: ⋆.

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
