# Peer review of "Delphi Method to Achieve Clinical Consensus for a BPMN Representation of the Central Venous Access Placement for Training Purposes"

_ijerph, 2020, doi:10.3390/ijerph17113889_

Round 1

Reviewer 1 Report

Dear authors,

your paper presents an approach to the consensual modelling of a clinical procedure using the Delphi model. This work is very well structured and the method is sound.

The process model in Figure 1 is too small (you can only read the text on screen after setting zoom to 400%). This should definitely be improved, even if it means, that you get a full-page model.

I understand that the model cannot cover the whole complexity of the procedure (especially with complications). However, I wonder if the steps that follow a poorly placed guidewire should be modelled as well (Guidewire in good position -> NO)? Would you really just start puncturing again, or is there a need for removing the old guidewire? Also for the check at the end - what would you do if the radiology method shows bad positioning?

Of course you can't change the agreed upon model, but I think you could provide a bit more details in the description. At the moment it is a "happy path" model. What do you think is necessary to make it exhaustive? Could it still be used for training then, or would it be too large to comprehend? Do you think that doctors would be willing to participate in a Delphi review of an exhaustive model?

Minor remarks:

  • Change survey to surveys (plural) in line 124.
  • The sentence starting at line 159 is long and hard to comprehend. Please consider splitting.
  • Could you provide the link to the survey or add the survey in the appendix? Some of the questions would be interesting for the audience, e.g., Part 4 and how physicians were able to answer (how did they describe where to add activities to the model?).

Your work is well presented and provides a valuable contribution.

Reviewer 2 Report

The paper aims at defining a procedural documentation for the central venous access placement. The problem of defining procedural documentation for medical procedures is well known in the literature, and the proposed solution, using BPMN, is a valuable one already adopted by several other approaches. Besides proposing a BPMN model for the central venous access placement, authors evaluate and improve it by using a Delphi method. The paper is well written and it technically sounds. Although the idea of using BPMN to model medical procedures is not new (as also stated by the authors), I think that the paper is a valuable piece of work providing a better (procedural) documentation of the central venous access placement procedure with respect to the state of the art.

Considerations:

  • Title: since the main result of this work is a BPMN model of the central venous access placement, I think that BPMN should be included in the title.
  • Page 1 line 14: “This approach is an alternative […]” I think that “alternative” is ambiguous. The procedural documentation (and knowledge) should be complementary with respect to the declarative one (i.e., the “classic” one).  
  • Page 2 line 69: the first BPMN model is not shown in the paper. This makes difficult to evaluate the effectiveness of the Delphi methodology used to improve it.
  • Page 5 Table 2: it would be useful to highlight (e.g., with different background colors) the activities removed from the model
  • Page 5 lines 144-151: I find interesting that some of alternative activities did not reach the inclusion thresholds. Have authors hypothesized a reason for that? For instance, does it depend on the way the question is asked to the experts? The fact that the activity “Identify vein with color Doppler” has a change of -14.58% from the first to the second survey is also curious. Authors should comment these aspects.
  • Page 6 Figure 1: even if BPMN is a well-known standard for process model, a brief description of the BPMN constructs used in the diagram would be useful for its comprehension by non-experts
  • Page 6 Figure 1: are the inclusive decisions (“and/or”) actually inclusive in the medical practice? I mean, I suppose that alternative activities are usually executed following a process by “trial and error”: the physician executes one of the alternative procedures, if the result is satisfactory, she continues to the next step, otherwise she tries with another alternative procedure. If this is the case, the BPMN model should be corrected.

Typos:

  • Page 2 line 75: “answer” should be “answers”

Reviewer 3 Report

Thank you for your interesting work. The paper presents a BPMN model for a medical procedure, which is assumed to have better outcomes and standardize practices in the medical industry.

Introduction:

The introduction provides a good overview of the topic. However, the motivation for this work can be better presented. Instead of just learning difficulties, the actual impact of unstandardized processes can be explained to motivate the work further.

Method:

For people who are not aware of BPM, it may not be a bad idea to mention about the BPM lifecycle.

For the development of initial model, I am not sure if using certain literature is sufficient. Checklist captures the end product rather than intermediate activities, which are required to obtain the final product. There can be a recommended way of obtaining the final product present in the checklist, which may be missing in the initial model. I feel the initial model should be derived based on insights from a hospital and augmented with literature than vice versa. Your explanation of Delphi method is good.

Results:

For a study like this it would have been good to see the progressive refinement in the BPMN model. It is difficult to understand what differences the Delphi study made without visual aspect. I understand space can be an issue, but maybe an external url would have been good.

I have another fundamental question. How many hospital staff are able to understand BPMN notation? That is something that needs to be understood as well. Maybe having the same model is a more familiar option may be a better approach. Alternatively, an evaluation showing that the practitioners found the BPMN model useful would be good. Is there a real difference between this model and checklist? Could the checklist be augmented based on the Delphi study. Introduction of anything new results in cognitive overload, and I feel these things need to be taken into consideration before proposing such a method, in other words, a feasibility study.

Other comments: 1. It would be good to place Table 2 in appendix and present the results in an innovative way. 2. Comparison of different roles of Delphi panel would be good.

Good luck with revisions.
